# The effect of dose on the antimalarial efficacy of artesunate-mefloquine against *Plasmodium falciparum* malaria: a protocol for systematic review and individual patient data (IPD) meta-analysis

Rashid Mansoor, [1,2] Prabin Dahal, [1,2] Georgina S Humphreys,[3] Philippe Guerin,[1,2] Elizabeth A Ashley,[2,4] Kasia Stepniewska[1,2]

RM and PD contributed equally.

For numbered affiliations see end of article.

**Correspondence to**
Dr Kasia Stepniewska;
kasia.stepniewska@wwarn.org

## ABSTRACT

**Introduction** Antimalarial posology based on weight bands leaves patients at the margins vulnerable to receiving either lower or higher weight-adjusted (mg/kg) dosages. This article aims to describe the protocol for systematic review and individual patient meta-analysis (MA) for a study of the distribution of artesunate and mefloquine dosage administered in patients with uncomplicated *Plasmodium falciparum* malaria treated with an artesunate-mefloquine (AS-MQ) regimen. Relationship between mg/kg dose and therapeutic outcomes will be assessed through a one-stage individual participant data (IPD) MA.

**Methods and analysis** Therapeutic efficacy studies with the AS-MQ regimen will be identified by searching the following databases: PUBMED, EMBASE and Web of Science. The corresponding authors of the relevant studies will be requested to share the IPD for the purpose of this MA to a secured repository. All available studies will be standardised using a common data management protocol and pooled into a single database. The relationship between mg/kg dosage and treatment failures will be assessed using a Cox regression model with study sites considered as a shared frailty term. Safety parameters will be explored where available.

**Ethics and dissemination** This IPD MA met the criteria for waiver of ethical review as defined by the Oxford Tropical Research Ethics Committee as the research consisted of secondary analysis of existing anonymous data. The results of this analysis will be disseminated at conferences, WorldWide Antimalarial Resistance Network website and any peer-reviewed publication arising will be made open source.

**PROSPERO registration number** CRD42018103776.

## INTRODUCTION

The combination artesunate-mefloquine (AS-MQ) was the first antimalarial regimen developed as an artemisinin-based combination therapy (ACT) when mefloquine

## Strengths and limitations of this study

► With the exception of recent studies in Southeast Asia, the regimen artesunate-mefloquine has consistently demonstrated an efficacy greater than 95% and treatment failures in any single antimalarial study are few. This individual participant data (IPD) meta-analysis (MA) will allow a robust exploration of host, parasite and drug factors associated with therapeutic outcomes, which otherwise would not be possible.

► The proposed IPD MA will allow exploration of variations in the weight-adjusted dosage received by patients, which is not possible with aggregate data MA.

► The IPD MA will be carried out as a study group under the auspices of the WorldWide Antimalarial Resistance Network, which has championed responsible data sharing and advocates translational research.

► A limitation of this analysis will be heterogeneity between studies included in terms of design, patient population and the susceptibility of the parasites against the drug regimen.

(MQ) resistance became rampant along the Thai–Myanmar border in the early 1990s.[1] The efficacy of a combination regimen (artemisinin derivative+partner component) depends on the ability of the partner component to mop up the residual parasites leftover after the initial and highly potent anti-parasitic activity of the artemisinin derivatives. This requires the dosage of the partner drug to be sufficient to ensure that blood concentrations exceed the minimum inhibitory concentration of the parasites until all the parasites have been killed. Manufacturers' recommendations regarding antimalarial posology are often pragmatic and the dose is

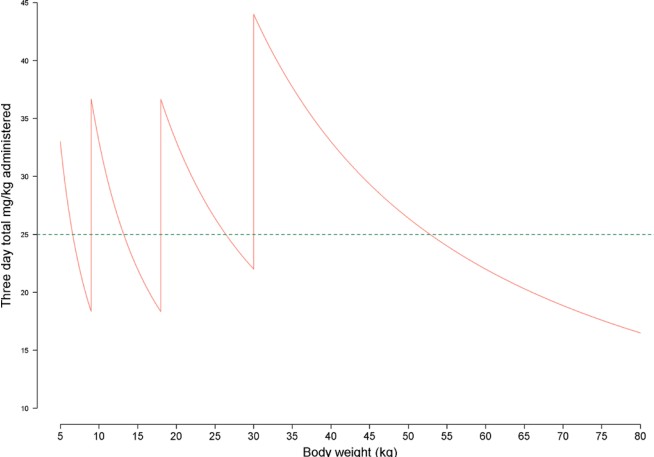

**Figure 1** mg/kg Dose variations for mefloquine (MQ). The WHO recommendation on the 3-day fixed-dose artesunate-mefloquine regimen is to administer once daily 50 mg of MQ for those weighing 5–8 kg, 100 mg for 9–17 kg, 200 mg for 18–29 kg and 400 mg for those who weigh ≥30 kg. The horizontal dotted line represents the target MQ dose of 25 mg/kg. Since the dosing is based on weight bands, this leads to the saw-tooth dosing curves.

administered based on weight 'banding'. This approach inevitably results in some patients at the band margins receiving either lower or higher dosages when adjusted for body weight (figure 1). Young children are particularly vulnerable to extreme total dosages especially when drug administration is based on tablets rather than paediatric formulations or suspensions. This may lead to sub-therapeutic drug concentrations in the blood plasma and such underexposure has been related to poorer therapeutic response for some of the widely used ACTs.[2–4]

Until recently ACTs have consistently demonstrated an efficacy greater than 95% and treatment failures in any single antimalarial study are few, thus limiting the ability to draw inferences regarding putative factors associated with therapeutic outcomes. Individual participant data (IPD) meta-analysis (MA) is being used increasingly to explore some of the putative factors which otherwise would not be possible through aggregate data MA.[5] Such an IPD MA approach has been used to assess the dose–response relationships for the ACT regimens of artesunate-amodiaquine, artemether-lumefantrine and dihydroartemisinin-piperaquine.[2 3 6] These studies have demonstrated that the drug formulation (fixed vs loose) and underexposure in the paediatric population due to weight banding are deterministic of poorer therapeutic outcomes. A thorough evaluation of the dose–response relationship for the regimen AS-MQ is lacking and this IPD MA aims to address this research gap.

## OBJECTIVES
The overall aim of this study is to determine the mg/kg dosing range of artesunate (AS) and MQ adopted in clinical trials and investigate the effects of mg/kg dosing on clinical outcome.

The specific objectives are:
► To investigate the effects of MQ and AS mg/kg dosing on early and late clinical outcomes (treatment success or failure).
► To investigate the tolerability of AS-MQ across different study sites, population and age groups.

## METHODS AND ANALYSES
### Criteria for study eligibility
Studies identified will be deemed eligible for the purpose of this analysis if they meet the following criteria:
► Prospective clinical efficacy study (defined as a trial which enrolled patients with confirmed diagnosis of malaria and who were follow-up for at least 28 days post-treatment) of uncomplicated *Plasmodium falciparum* (either alone or mixed infections) in patients of all ages.
► Assessing the efficacy of a fixed-dose AS-MQ combination, either as single tablet type, or co-blister pack of more than one tablet type, or assessing the efficacy of a loose combination of AS-MQ.
► Where AS was given over 3 days (with any number of doses per day) with a target total dose of 6–30 mg/kg.
► Where MQ was given over 1–3 days, on any of days 1–3 (with any number of doses per day) with a target total dose of 15–33 mg/kg.
► Where all AS and MQ were administered orally.
► With a minimum of 28 days follow-up.
► With genotyping performed for late parasite recurrence.
► With individual patient data on dosage of MQ received (actual or per protocol) by patients (dosage per tablets, number of tablets given per dose and duration of treatment).

### Criteria for study exclusion
► Where other antimalarial drugs were given in addition to the initial AS-MQ treatment regimen, except for a single dose of primaquine of 0.25 mg/kg in the first 3 days.

### Types of study participants
Patients with uncomplicated *P. falciparum* malaria will be included in this IPD MA. The following patients will be excluded from the analysis:
i. Severe *P. falciparum* malaria.
ii. Pregnant women.

### Types of intervention/exposure and controls
► Fixed-dose combination of AS-MQ, either as single tablet type, or co-blister pack of more than one tablet type, or loose combination of AS-MQ. AS given over 3 days (with any number of doses per day) with a target total dose of 6–30 mg/kg. MQ given over 1–3 days, on any of days 1–3 (with any number of doses per day) with a target total dose of 15–33 mg/kg.

### Types of outcomes
i. Parasitological and clinical efficacy.

ii.   Adverse events.

### Information sources and search strategy

We will carry out a systematic review and search PubMed, EMBASE and Web of Science to identify publications with AS-MQ between January 1990 and July 2018; the full search terms are available from the PROSPERO registration and also presented as supplementary file 1. We have not planned to search grey literature for the purpose of this review.

Any important protocol amendments will be documented in the PROSPERO registration. Studies will be included regardless of language and publication status. Study screenings will be carried by two independent reviewers who will screen title, abstract, full text as necessary. The following studies will be excluded: animal models (eg, mouse malarias *Plasmodium berghei, Plasmodium chabaudi*), publications only including severe malaria, studies with follow-up <28 days, data previously included in another published study, prophylaxis or mass drug administration studies, studies in healthy volunteers/challenge studies, or studies in asymptomatic patients or pregnant women.

### Data acquisition and data management
#### Collating IPD

Principal investigators of the published (or unpublished) studies identified from the literature search will be invited to share IPD. At least three emails will be sent out in case of non-response. Researchers agreeing to the terms and conditions of the submission will be requested to upload anonymised IPD to the WorldWide Antimalarial Resistance Network (WWARN) repository through a secure web portal. Figure 2 shows the process map which depicts the different phases of data procurement:

data acquisition from the contributors, data standardisation and their subsequent reuse in IPD MA.

Data will be fully anonymised and handled in compliance with the UK Data Protection Act to protect personal information and patient privacy. Original data will be stored on a secure server hosted by the University of Oxford.

### Data management

Raw data from individual studies will be standardised using an open and transparent data management and statistical analysis protocol.[7] Investigators will be further contacted for validation or clarification, if required, and individual study protocols will be requested. On standardisation, the data will be stored in a relational database of several tables containing information on drug regimen, parasitological, clinical, and haematological assessments, genotyping and therapeutic outcomes, all linked by a unique patient identifier.

### Data contributors' participation

All the researchers who share individual patient data from eligible studies will become part of the study group; will have an opportunity to contribute to the analysis, interpretation of the results, manuscript preparation; and will be listed as coauthors on the publication arising from these analyses according to the WWARN publication policy.

### Statistical analysis plan
#### Study population

The following patients will be included in the analysis:

i.   Information is available on drug dosage, either as exact number of tablets received, exact mg/kg dose received or number of tablets planned per protocol.

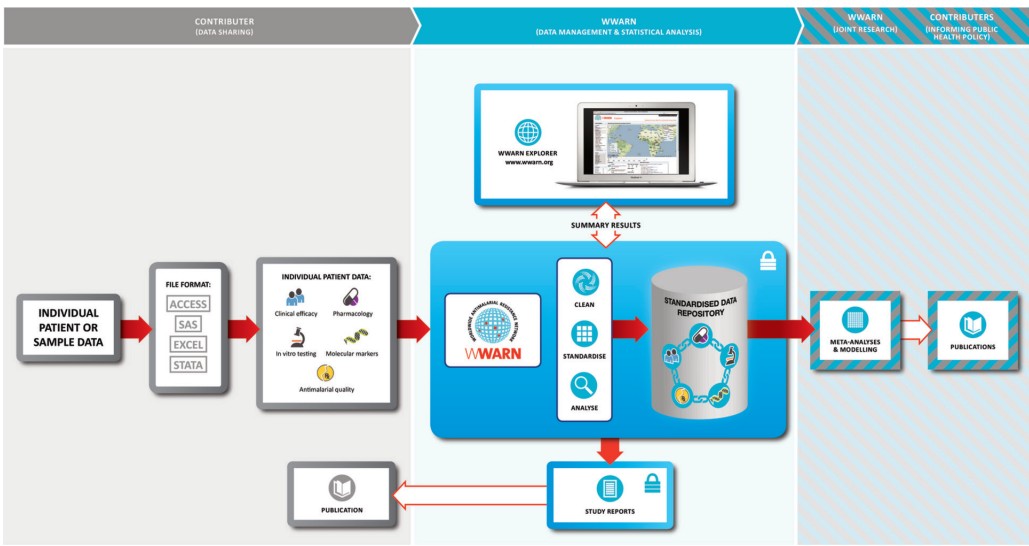

**Figure 2** Data acquisition and standardisation process utilising the WorldWide Antimalarial Resistance Network (WWARN) database. Individual participant data are shared by researchers to the WWARN secured portal (left panel). The studies are standardised using a common WWARN protocol and the tables of clinical, parasitological and drug measurements are stored in a secured repository with relational database (middle panel). On standardisation, the data set is shared back with the study investigator, and subsequently used in meta-analyses to answer questions of public health importance (right panel).

ii. Date of the last day of follow-up or length of follow-up.

The following patients will be excluded from the analysis:

i. Received other antimalarial drugs during follow-up before recorded *P. falciparum* treatment failure.

ii. Results of genotyping performed for late parasitological outcome are not available.

iii. Missing confirmation of *P. falciparum* infection on enrolment.

iv. Missing age or weight or gender.

v. Other deviations, as defined in the data management plan[7]:

 a. Haemoglobin (Hb) <50 g/L on day 0.

 b. Haematocrit (Ht) <15% on day 0.

## Outcomes

Primary: PCR-corrected *P. falciparum* recrudescence.

Secondary: PCR-corrected *P. falciparum* reinfection.

PCR-uncorrected *P. falciparum* recurrence.

Early parasitological responses on days 1, 2 and 3.

Gametocyte carriage within 14 days of treatment initiation in patients without gametocytaemia at enrolment.

Anaemia status on day 7.

Adverse symptoms developed after the drug administration.

The primary endpoint for this IPD MA is PCR genotyping corrected risk of *P. falciparum* recrudescence (treatment failure) on day 42. Day 42 was selected as the primary endpoint based on the current recommendations from the WHO as outlined in the 2009 protocol, which suggests that day 42 be the minimum follow-up period for the MQ regimen.[8] In the analysis of the primary endpoint, patients in whom new infections are observed during the study follow-up, or those who are lost to follow-up, will be censored; the former on the day new infection was observed and the latter on their last recorded visit day. For the analysis of PCR-corrected new infections, patients with recrudescence and those who are lost to follow-up will be censored. Further definitions of status and other censorship are detailed in the clinical module Data Management and Statistical Analysis Plan (DMSAP).[7]

Acute drug vomiting within an hour of treatment administration, general vomiting within 7 days of treatment initiation, diarrhoea within 7 days of treatment initiation and neuropsychiatric adverse events (where available) are secondary endpoints.

## Variables and definition

AS and MQ doses received will be calculated from the number of tablets administered to each patient daily. If the actual number of tablets received is not recorded, the total dose in mg or mg/kg recorded as administered to each patient will be used. If none of these are available, administration as per protocol will be assumed. The current recommended daily MQ dose is 8.3 (range 5–11) mg/kg.[9] A patient will be classified as underdosed if the 3-day total mg/kg MQ dose is less than 15 mg/kg.

Nutritional status in children under 5 years of age will be assessed using standardised age, weight, height and gender specific growth reference standards according to the WHO 2006 recommendations using igrowup Stata package.[10] Anthropometric indicators include weight-for-age z-score (WAZ), height-for-age z-score and weight-for-height z-score. The nutritional status of a child will be given as a z-score and classified as stunted, underweight or wasted as defined in the WHO guidelines.

The falciparum malaria transmission intensity of the study sites will be assessed using the prevalence estimates generated by the Malaria Atlas Project based on the latitude, longitude and the year the study was conducted.[11 12]

Anaemia will be defined as Hb <100 g/L or Ht <30%. Severe anaemia is defined as Hb <70 g/L or Ht <20%. Fever will be defined as body temperature >37.5°C.

Parasite resistance status will be defined (data permitting) for each patient from Southeast Asia region based on the reported prevalence of mutations of molecular markers (pfmdr1, kelch13) or the distribution of parasite clearance half-life for their study site and year of admission.[13 14] For other locations, we will assume that parasites are sensitive to the AS-MQ regimen.

## Descriptive summary

### Summary of the studies

Summary of included studies will be presented with respect to study location, years of study, study population, duration of follow-up, AS-MQ drug formulation, methodology for parasite quantification, methodology for PCR genotyping and supervision of drug administration. PCR-corrected/uncorrected outcomes will be used to compute the Kaplan-Meier (K-M) estimates using the censoring rules outlined earlier. The K-M estimates will be presented graphically together with the number of patients in the risk set.

### Summary of the patients

Summary of baseline characteristics of the patients included in the analysis will be presented for each study, by region and in overall. The following baseline characteristics of patients will be presented: age; weight; parasitaemia on enrolment; presence of fever (body temperature >37.5° C); Hb (or Ht); anaemia (Hb <100 g/L) or severe anaemia (Hb <70 g/L); gametocytes on presentation; description of infection (*P. falciparum* or mixed infections); total mg/kg dose for each drug component; dosing strategies (age-based, weight-based, and so on); dose formulation (fixed or loose) and supervision of drug administration. The number of available patients will be summarised for all variables, proportion will be used for categorical or binary variables and mean and standard deviation (or median and range) will be used for continuous variables.

### Analysis of the primary endpoint

The efficacy estimates for each of the studies will be summarised using the K-M method.

Cox regression analysis will be carried out to identify the predictors associated with parasitic recrudescence using a one-stage IPD MA. Random effects in the form of shared gamma frailty parameters will be used to adjust for study-site effect and account for unobserved statistical heterogeneity.[15] Schoenfeld residuals against transformed time will be used to determine if the assumption of proportional hazard is met. Cox-Snell residuals will be examined to determine the appropriateness of model fit. Martingale residuals will be used to assess the functional form of the covariates. Potential non-linear relationships between continuous variable and the treatment outcomes will be investigated using multivariable fractional polynomials.[16] If the assumption of proportional hazards is not satisfied, alternative approaches such as piecewise proportional hazards model, interaction with time, stratifying by the variable for which the assumption is not satisfied or flexible parametric models will be considered. Variable selection process will follow a procedure described below.

### Analyses of secondary endpoints
#### *P. falciparum* new infection
The analysis of new infections will be similar to the analysis of the primary endpoint.

#### Parasite clearance
Early parasitological responses will be assessed by the parasite positivity rate, which is the proportion of patients remaining parasitaemic on days 1, 2 and 3 post-treatment administration.[17] The relationship between mg/kg dosage of the AS and MQ on early parasitological responses will be explored using a logistic regression model with study sites fitted as random effect. Variable selection and additional sensitivity analyses will follow the plan as outlined for the primary endpoint.

#### Gametocyte carriage
Gametocyte carriage during the study follow-up will be stratified by the gametocytaemia status at baseline. For those with documented gametocytaemia at enrolment, proportion of patients in whom gametocyte has cleared will be reported. For those without gametocytes on enrolment, the proportion of patients in whom gametocytes have evolved will be presented. The relationship between mg/kg dosage of the AS and MQ on gametocyte endpoints will be explored using a logistic regression model with study sites fitted as random effect. Variable selection and additional sensitivity analyses will follow the plan as outlined for the primary endpoint.

#### Haematological insult
Anaemia during the study follow-up will be stratified by the anaemia status at baseline. For those who are anaemic at enrolment, the proportion of patients who have recovered will be reported. For those who are not anaemic at enrolment, the proportion of patients whom are subsequently anaemic will be presented. The relationship between mg/kg dosage of the AS and MQ on anaemia endpoints will be explored using a logistic regression

model with study sites fitted as random effect. Variable selection and additional sensitivity analyses will follow the plan as outlined for the primary endpoint.

### Safety endpoints
The proportion of patients with acute drug vomiting, vomiting and diarrhoea, neuropsychiatric adverse events within a week of treatment initiation will be reported. The relationship between the mg/kg MQ dose and safety endpoints will be evaluated using a logistic regression model with study sites fitted as a random effect, if data permit. Variable selection and additional sensitivity analyses will follow the plan as outlined for the primary endpoint.

### Variable selection strategy
The following covariates will be examined: age, sex, weight, baseline parasitaemia (except for new infection analysis), WAZ, underweight for age termed underweight (defined as WAZ < −2), Hb, gametocytes on presentation (except for new infection analysis), history of malaria (if available); description of infection: mixed species infections (except for new infection analysis), presence of markers of drug resistance, eg kelch13 mutations or pfmdr1 amplification (if available), details of treatment received: total mg/kg dose of AS and MQ, regimen, drug supervision and vomiting of medication. Year of enrolment will also be included to account for changes in parasite susceptibility over time.

A general strategy recommended by Collet (2015)[18] will be followed for the construction of multivariable regression model:

i.   All possible risk factors will be examined in a univariable analysis. The log-likelihood estimates $(−2 \times Log\hat{L})$ will be compared against the null model to assess if any of the variables reduces its value at 5% level of statistical significance.

ii.  All the variables identified in step (i) will be fitted together in one model and variables that are not significant in the presence of other variables based on the results of the Wald test will be identified.

iii. A likelihood ratio test (LRT) will be used to assess the impact of omitting variables identified in step (ii). If the omitted variable does not significantly impact the model log-likelihood, then they will be dropped. Only those variables which lead to significant change in log-likelihood are retained.

iv.  All variables excluded from step (i) will be added to the model identified in step (iii) one by one to check if they provide any improvement to the model.

v.   A final check of the model identified in step (iv) will be carried out to ensure that none of the variables in the model can be omitted without significantly increasing the model log-likelihood, and none of the excluded variables significantly reduce the model log-likelihood.

Comparison of likelihood between nested models will be conducted using LRT. Akaike's information criterion will be used to compare non-nested models. Treatment

dosage, drug formulation and baseline parasitaemia will be included in the multivariable model as a priori forced variables regardless of their statistical significance. Variables with more than 50% observations missing will not be included in multivariable analysis. Interactions will be assessed between dosing and the following variables: region, age group, transmission intensity, hyperparasitaemia (parasitaemia >100 000 parasites/µl), date of enrollment.

### Assessment of statistical heterogeneity

The multilevel logistic or Cox models will be used for explaining study-site heterogeneity. Heterogeneity across study sites will be assessed by the variance of the shared frailty term estimated in the random effect Cox model or variance of the random intercepts in logistic regression. In addition, intraclass correlation in logistic regression model will be reported.

### Subgroup analyses

Analyses will be conducted by geographical region, drug regimen and resistance status if data permit.

### Sensitivity analyses

A model will be refitted with each study's data excluded, one at a time, and a coefficient of variation around the parameter estimates will be calculated. This would identify any influential studies, that is, studies with unusual results (due to variations in methodology, patient population and so on) that affect the overall pooled analysis findings. To assess the impact of missing data (covariates, PCR genotyping results), sensitivity analysis will be performed to see if our main conclusion is affected or not by the exclusion of patients with missing data. Multiple imputation (MI) will be used for handling missing data for missing covariates and missing outcomes. MI will be carried out for covariates with missing proportion less than 50%.[19]

### Quality assessment/risk of bias assessment in studies included

Two reviewers will independently assess risk of bias. The risk of bias within and across the studies included in the analysis will be carried out using the GRADE guidelines.[20] Cochrane risk of bias tool 2.0 will be used to assess risk of bias in individual randomised controlled trials. Publication bias will be assessed by a funnel plot.[21]

### Assessment of risk of potential bias in missing studies

Despite best possible efforts, it is anticipated that raw data from all the identified studies will not be available. Risk of potential bias in these studies will be assessed using a two-stage IPD MA for the reported efficacy outcomes.

### Further development of statistical analysis plan

The main analysis is planned as described earlier. Modification or additional analyses may be required as the data collection progresses. An updated statistical analysis plan will be available on the WWARN study group website.[22]

### Software

All statistical analyses will be carried out using R (The R Foundation for Statistical Computing) or StataM 15. Alternative statistical software will not require amendment of this SAP.

### Ethics and dissemination

This IPD MA met the criteria for waiver of ethical review as defined by the Oxford Tropical Research Ethics Committee as the research consisted of secondary analysis of existing anonymous data.[23] All studies included in this analysis received local ethical approvals and our pooled IPD MA will be addressing scientific questions that are very similar to the original research questions.

Findings will be reported following the PRISMA-IPD guidelines[24] at open-access peer-reviewed journals. This systematic literature review and IPD meta-analysis is registered to PROSPERO and this protocol has been reported following the PRISMA-P guidelines.[25] Any publications based on the findings of this IPD MA will be in accordance with the guidelines of the International Committee of Medical Journal Editors.

### Patient and public involvement

Patients were not involved in the development of the research question, outcome measure or study design.

## DISCUSSION

Large-scale deployment of highly efficacious ACT regimens such as AS-MQ has been the cornerstone of global malaria control for over a decade and this has contributed to the global reduction of mortality and morbidity associated with malaria.[12] Maintaining these gains is highly dependent on efficient health systems, sustainable global funding and the current status of antimalarial drug resistance. The 2017 WHO report found that globally the number of malaria cases has stopped dropping and mortality has crept up compared with 2016, suggesting that the recent public health gains remain fragile.[26 27] To make things worse, a health calamity is looming large due to the emergence of resistance to artemisinins in Southeast Asia which is threatening to reverse the remarkable progress achieved over the past decade.[28] In the absence of an alternative treatment class to replace the ACTs as first line therapy, preserving the currently available drugs remains the top-most priority and this requires the highest form of evidence regarding the susceptibility of the parasites against the antimalarial drugs. AS-MQ retains extremely high efficacy in most locations except Thailand, and in any given trial, only few treatment failures have been observed which limits the utility of any single study in answering questions regarding the dose–response relationship. IPD MA provides an alternative strategy.

IPD MA is now considered the gold standard for evidence synthesis and allows exploration of different risk factors which otherwise would not be possible through the aggregate data MA.[5] This IPD MA is designed to

explore the variability in drug dosage administered in patients with uncomplicated *P. falciparum* malaria, treated with AS-MQ. The WHO-recommended AS-MQ regimen is administered as a 3-day course, with a total of 12 mg/kg of AS and 25 mg/kg of MQ split over 3 days. Due to the poor tolerability of high-dose MQ, the dose of MQ is usually divided into either two doses (15 and 10 mg/kg), or three as a fixed-dose combination (8 mg/kg/day). The fixed-dose combination has been shown to provide better efficacy and improve treatment adherence for artesunate-amodiaquine.[6] Such a comparison is yet to be made for the AS-MQ regimen, and in this IPD MA we propose to compare the fixed and loose formulations of the regimen with regard to the drug dosing, tolerability, efficacy and practicality of the dose banding.

In conclusion, this pooled analysis will provide critical information regarding the relationship between drug dosage and parasitological responses post-treatment with AS-MQ. The assessment of the host, parasite and drug determinants that influence the treatment response can provide evidence-based guidance for monitoring the early signs of artemisinin resistance and effective case management that will be critical in optimising malaria control and containment efforts.

**Author affiliations**
[1]WorldWide Antimalarial Resistance Network (WWARN), Oxford, UK
[2]Centre for Tropical Medicine and Global Health, Nuffield Department of Medicine, University of Oxford, Oxford, UK
[3]Wellcome Trust, London, UK
[4]Myanmar-Oxford Clinical Research Unit (MOCRU), Yangon, Myanmar

**Acknowledgements** We would like to thank Dr Makoto Saito for his several helpful suggestions. We would like to also thank Brittany Maguire for her help with the manuscript.

**Contributors** RM, PD, PG, EAA and KS: conceived the idea and wrote the first draft of the protocol. GSH: systematic review of all published antimalarial studies. GSH, EAA and KS: data acquisition and standardisation. All authors read and approved the submission of the final draft of the study protocol.

**Funding** The WorldWide Antimalarial Resistance Network (RM, PD, GSH, PG and KS) is funded by a Bill and Melinda Gates Foundation grant and the ExxonMobil Foundation. The funders did not participate in the study development, the writing of the paper, decision to publish or preparation of the manuscript.

**Competing interests** None declared.

**Patient consent for publication** Not required.

**Ethics approval** The individual patient data meta-analysis met the criteria for waiver of ethical review as defined by the Oxford Tropical Research Ethics Committee since the research consisted of secondary analysis of existing anonymous data. Each study included in the analysis received local ethics approvals.

**Provenance and peer review** Not commissioned; externally peer reviewed.

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
