## [Reviewer comments · BMJ Open]

ARTICLE DETAILS

TITLE (PROVISIONAL)	The effect of dose on the antimalarial efficacy of artesunate-mefloquine against Plasmodium falciparum malaria: a protocol for systematic review and individual patient data (IPD) meta-analysis
AUTHORS	Mansoor, Rashid; Dahal, Prabin; Humphreys, Georgina; Guerin, Philippe; Ashley, Elizabeth; Stepniewska, Kasia

VERSION 1 - REVIEW

REVIEWER	Henry Maia Peixoto - University of Brasilia - Brazil.
REVIEW RETURNED	05-Dec-2018

GENERAL COMMENTS	“The effect of dose on the antimalarial efficacy of artesunate-mefloquine against Plasmodium falciparum malaria: a protocol for systematic review and individual patient data (IPD) meta-analysis”. Mansoor et al. The authors have produced a very interesting protocol for systematic review and individual patient data meta-analysis. There is not much to add except for a few comments. Abstract and introduction - I suggest the authors change the phrase “This study aims to describe the distribution of artesunate and mefloquine dosage administered in patients with uncomplicated P. falciparum malaria treated with an artesunate-mefloquine regimen”. The authors should point out that the paper aims to describe the protocol for a systematic review and individual patient data meta-analysis. Methodology - I would suggest that the authors consider the relative importance of outcomes according to the GRADE approach. Available in: (https://gdt.gradepro.org/app/handbook/handbook.html). An example can be found in Peixoto et al. Efficacy and safety of artesunate-mefloquine therapy for treating uncomplicated Plasmodium falciparum malaria: systematic review and meta-analysis. Available in https://academic.oup.com/trstmh/article/110/11/626/2758632 - The authors should add full search terms in the body of the paper or in the supplementary material.
--

REVIEWER	sharon straus university of toronto, canada
REVIEW RETURNED	18-Dec-2018

GENERAL COMMENTS	congratulations to the authors for developing this protocol for an IPD MA. some suggestions to further strengthen their manuscript:  1. on page 6, please define prospective clinical efficacy study 2. on page 6 - add page age for eligibility criteria 3. outcomes - will they assess death? hospital/er admissions? 4. page 7 - was the search done by an experienced librarian? was it appraised using PRESS? will a grey lit search be done? will references of included studies be checked for additional studies? 5. feasibility of obtaining ipd - what will they do if they don't obtain any ipd? and if they do obtain it, will they invite the authors of those studies to review the analyses etc? 6. consider providing a reference for assessment of publication bias 7. will a sensitivity analysis be done for risk of bias?
---

VERSION 1 – AUTHOR RESPONSE

#Reviewer: 1: Henry Maia Peixoto

Reviewer 1: The authors have produced a very interesting protocol for systematic review and individual patient data meta-analysis. There is not much to add except for a few comments.

Abstract and introduction: I suggest the authors change the phrase “This study aims to describe the distribution of artesunate and mefloquine dosage administered in patients with uncomplicated P. falciparum malaria treated with an artesunate-mefloquine regimen”. The authors should point out that the paper aims to describe the protocol for a systematic review and individual patient data meta-analysis.

Author’s response: Thank you for the interest in our protocol. We agree with the reviewer’s suggestion and have made changes in the abstract.

Reviewer 1: Methodology - I would suggest that the authors consider the relative importance of outcomes according to the GRADE approach. Available in: (<https://gdt.gradepro.org/app/handbook/handbook.html>). An example can be found in Peixoto et al. Efficacy and safety of artesunate-mefloquine therapy for treating uncomplicated Plasmodium falciparum malaria: systematic review and meta-analysis. Available in <https://academic.oup.com/trstmh/article/110/11/626/2758632>

Author’s response: We thank Reviewer 1 for this comment and note the approach referenced in Peixoto et al (2016). We agree with the reviewer that day 63 outcome should be considered critical for the recommendation. The rationale being antimalarial drug efficacy is inevitably affected by the terminal elimination half-life of the partner component. Mefloquine has a long half-life thus suppresses the parasites for a far longer time than drugs with much shorter elimination half-life. However, for this

systematic review IPD-MA protocol, we have defined day 42 as the primary endpoint based on the current recommendations from the WHO as outlined in the 2009 protocol, which suggests that day 42 be the minimum follow-up period for the mefloquine regimen. Changes have been made to lines 205-208 in the main text. The following text verbatim from the WHO 2009 protocol:

“Thus, as a compromise, a 28-day follow-up is recommended as the minimum standard to allow national malaria control programmes to capture most failures with most medicines, except mefloquine and piperaquine, for which the minimum follow-up should be 42 days (Stepniewska et al., 2004)

”https://apps.who.int/iris/bitstream/handle/10665/44048/9789241597531_eng.pdf;jsessionid=1A36840004E5EC9111503F8A0A4AE483?sequence=1

Access date: 14/03/2019

Reviewer 1: The authors should add full search terms in the body of the paper or in the supplementary material.

Author’s response: We have now created a supplementary file listing the search strategy used for this systematic review.

#Reviewer: 2: Sharon Straus

Reviewer 2: Congratulations to the authors for developing this protocol for an IPD MA. Some suggestions to further strengthen their manuscript: on page 6, please define prospective clinical efficacy study

Author’s response: We would like to thank the reviewer for the kind appreciation. By prospective clinical efficacy study, we mean a trial which enrolled patients with confirmed diagnosis of malaria and who were followed-up for at least 28 days post-treatment as opposed to retrospective studies such as a case-control design in which a group of patients with a certain disease condition are asked about their exposure status. The text in the manuscript has been clarified (lines 108-110).

Reviewer 2: on page 6 - add page age for eligibility criteria

Author’s response: Thank you for pointing this out. Age was not part of the eligibility criteria as the disease affects population of all ages. We have now added “patients of all ages” as a criterion in line 111.

Reviewer 2: outcomes - will they assess death? Hospital/er admissions?

Author’s response: Death is an extremely rare outcome in an antimalarial study in uncomplicated falciparum malaria. Hence, we won’t be assessing death, or hospitalisation status of the patients.

Reviewer 2: page 7 - was the search done by an experienced librarian? was it appraised using PRESS? will a grey lit search be done? will references of included studies be checked for additional studies?

Author's response: An experienced reviewer (RB, see list of team members on the PROSPERO registration) developed the search terminology and tested the search against a number of eligible studies. PRESS has not been used. We intend to get our search strategy evaluated by an experienced librarian at Oxford University (possibly using PRESS), however this will depend on available resources so we have not added this in the protocol.

We do not plan to search grey literature for the purpose of this systematic review. This has been now added in lines 146-147 of the manuscript.

Reviewer 2: feasibility of obtaining ipd - what will they do if they don't obtain any ipd? and if they do obtain it, will they invite the authors of those studies to review the analyses etc?

Author's response: WorldWide Antimalarial Resistance Network (WWARN) has over 10 years of experience working on IPD meta-analyses in malaria and have collaborated with over 200 researchers (Humphreys, 2019) who shared their data for the purpose of those analyses in the past. Therefore, we are positive that data from the significant number of studies eligible for inclusion will be shared. For the remaining studies where the IPD is not available, we will consider them in our assessment of risk of bias, as described in lines 356-360.

All researchers who share their IPD will become part of the study group, will have an opportunity to contribute to the analysis and interpretation of the results, and will be listed as co-authors on the publication arising from these analyses according to the WWARN publication policy. This has now been clarified in the protocol, lines 174-178.

Humphreys GS, Tinto H, Barnes KI. Strength in Numbers: The WWARN Case Study of Purpose-Driven Data Sharing. Am J Trop Med Hyg. 2019 100(1):13-15

Reviewer 2: consider providing a reference for assessment of publication bias

Author's response: A reference for the funnel plot has now been added.

Reviewer 2: will a sensitivity analysis be done for risk of bias?

Author's response: We don't plan a sensitivity analysis specifically for the assessment of risk of bias.

VERSION 2 – REVIEW

REVIEWER	Henry Maia Peixoto Universidade de Brasília, Brazil
REVIEW RETURNED	03-Apr-2019

GENERAL COMMENTS	Dear Editor,
--------------

	The authors have produced a very interesting protocol for systematic review and individual patient data (IPD) meta-analysis. I recommend this paper for publication. Sincerely, Henry Peixoto
--	---